# Training and Cross-Validating Machine Learning Pipelines with Limited Memory

Martin Hirzel[1]  Kiran Kate[1]  Louis Mandel[1]  Avraham Shinnar[1]

[1]IBM Research

**Abstract**  While automated machine learning (AutoML) can save human labor in finding well-performing pipelines, it often suffers from two problems: overfitting and using excessive resources. Unfortunately, the solutions are often at odds: cross-validation helps reduce overfitting at the expense of more resources; conversely, preprocessing on a separate compute cluster and then cross-validating only the final predictor saves resources at the expense of more overfitting. This paper shows how to train and cross-validate entire pipelines on a single moderate machine with limited memory by using monoids, which are associative, thus providing a flexible way for handling large data one batch at a time. To facilitate AutoML, our approach is designed to support the common sklearn APIs used by many AutoML systems for pipelines, training, cross-validation, and several operators. Abstracted behind those APIs, our approach uses task graphs to extend the benefits of monoids from operators to pipelines, and provides a dual-backend implementation. Overall, our approach lets users train and cross-validate pipelines on simple and inexpensive compute infrastructure.

## 1 Introduction

This paper tackles the problem of training and cross-validating machine-learning pipelines on large datasets while only needing limited memory. This enables the use of a moderately-sized cloud compute node, thus avoiding the cost of upgrading to a more powerful machine and the complexity of upgrading to a cluster. It even lets data scientists use a laptop, thus simplifying debugging, avoiding data transfer, and reducing competition for compute resources.

Automated machine learning (AutoML) can jointly select algorithms and tune hyperparameters across an entire pipeline, including both data preprocessing and modeling [18, 46]. A *pipeline* is a directed acyclic graph of operators, where *operators* are data transformers (e.g., encoders, scalers) and predictors (e.g., classifiers, regressors). A pipeline *edge* $P \rightarrow S$ means that intermediate data transformed by a predecessor operator $P$ becomes the input to a successor operator $S$. As the most popular library for defining such pipelines today is sklearn [8], this paper adopts sklearn terminology, but its concepts apply more broadly. Many AutoML systems expect sklearn APIs [16, 18, 25, 36], and many third-party libraries conform to sklearn APIs [5, 10, 9, 14, 21, 24, 28]. We have extended the sklearn-compatible AutoML framework Lale [4] with our operators and algorithms.

Operators have two execution modes, training (`fit` in sklearn) and application (`transform` or `predict`). Training a pipeline requires both execution modes. To train an operator $S$ of a pipeline, the input data of $S$ must be transformed by its predecessors, which in turn must be already trained. Figure 1 shows a naive pipeline training algorithm. Here, *out* maps from operators to their intermediate transformed collection of output data, as well as the input data at the start of the pipeline. Unfortunately, this naive algorithm uses excessive memory, since it retains all non-batched intermediate data in *out*.

```
1  for each operator S in pipeline.topological_order:
2      S.fit(out[preds of S])
3      out[S] ← S.transform(out[preds of S])
```

Figure 1: Naive non-batched training algorithm.

One approach that avoids holding the entire training data in memory is deep learning [27]. Unlike general machine learning, deep learning uses an execution regime we call *partial transform*:

a partially-fitted predecessor layer *P* transforms data for a successor layer *S* so a batch passes all the way through the network without the predecessor being fully trained first. Unfortunately, partial transform does not work for all data preparation operators. For instance, the `OneHotEncoder` operator encodes categorical features as a one-hot numeric array; training this operator on additional batches can discover new categories, thus invalidating encodings of earlier batches. Hence, data preprocessing for deep learning usually happens separately, not in a single combined pipeline.

To enable training and cross-validating sklearn-style pipelines on large data without requiring partial transform, this paper uses *monoids* to support batching at the pipeline level. A monoid is simply a set with a binary associative *combine* operation and an identity element [7, 23, 45]. We express the training of several common operators as monoids, so they can train on each batch independently before combining learned coefficients from different batches. Next, we express the training of a pipeline of operators using a *task graph* with tasks for lifting batches to monoids, combining monoids, and applying operators. Using monoids in task graphs maximizes flexibility for reuse and scheduling, enabling us to minimize spilling to disk via a resource-aware schedule. Specifically, the associativity of monoids allows batches to be processed and combined in a more efficient order while guaranteeing equivalent end results to those of doing the work sequentially. Importantly for AutoML, monoids even let us eliminate some spurious work during cross-validation. This is because the associativity of monoids lets us do some computations only once per fold and then combine these reusable partial results into multiple sets of folds.

Our task-graph framework is general enough to faithfully emulate common machine-learning pipeline semantics. We express our monoidal operators via relational algebra to decouple them from backends. We implement the relational algebra operators on two backends, pandas [30] and Spark SQL [2]. The Spark SQL backend scales to large data and is distributed. The pandas backend also scales to large data thanks to our batching and spilling implementation but is not distributed and has lower framework overhead. This paper makes the following novel contributions:

- A monoidal framework to express the training of data preprocessing operators (Section 2).
- Algorithms for batched training or cross-validation of pipelines via task graphs that leverage monoids for resource-aware scheduling and spilling (Section 3).
- A generic implementation of the approach on top of relational algebra operators and two backends targeting pandas and Spark SQL (Section 4).

Our implementation is open-source: https://github.com/IBM/lale/tree/master/lale/lib/rasl.

## 2 Monoids

To train in limited memory, we want batch-wise training, but unlike in deep learning, we cannot pass a batch through the entire pipeline. Instead, we need to hold on to intermediate tranformed data whose size can exceed the memory budget, necessitating spilling. Since spilling is expensive, we want to minimize it by processing batches as independently as possible, without "harmful" [44] sequential dependencies on their predecessor. Fortunately, there is a very simple abstraction that supports exactly that: a *monoid*, which is a set with an associative operator. Besides enabling flexible schedules, monoids also enable us to eliminate some duplicate effort in cross-validation.

**Operators as Monoids**. A core insight of this paper is that several common machine-learning operators are *monoidal*, which we define as being expressible via the following three operations:

- `to_monoid : BatchXy → Monoid`, for lifting a batch to the monoid of the operator's learned coefficients;
- `combine : Monoid × Monoid → Monoid`, for combining learned coefficients from different batches; and
- `from_monoid : Monoid → TrainedOp`, for lowering learned coefficients from their monoidal form into a form suitable for the operator to be applied.

We express the training (`fit` in sklearn) of a monoidal machine-learning operator by applying the `to_monoid` operation to each batch and then applying the `combine` operation. And we express the application (`transform` or `predict`) by applying the `from_monoid` operation and using its results.

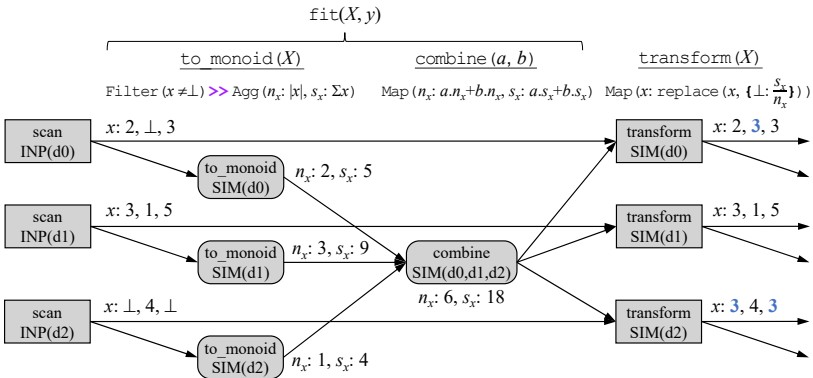

Figure 2: Monoid example: imputing missing values with mean (INP = input, SIM = SimpleImputer).

Figure 2 gives an example of imputing missing values with mean. It shows a part of a task graph for an input comprising three batches $d_0$, $d_1$, and $d_2$. The edges coming out of scan tasks are annotated with the values for an example column, $x$, that contains either numbers or a missing value, denoted $\perp$. The to_monoid operation filters non-missing values and computes their count $n_x$ and sum $s_x$. (The >> combinator denotes function composition.) The edges coming out of to_monoid tasks are annotated with combinable learned coefficients. For instance, in $x : 2, \perp, 3$, the number of non-missing values is $n_x : 2$ and their sum is $s_x : 5$. The combine task applies the monoid's operation; being associative, it can freely choose how to arrange the binary operations to combine multiple (here three) pieces. In this case, combine sums $n_x$ and $s_x$ from its predecessors. Finally, transform uses from_monoid to compute the arithmetic mean $\frac{s_x}{n_x}$, here $\frac{18}{6} = 3$, and replaces missing values with that number. For illustration purposes, the example shows only 3 rows, 1 column, and 3 batches; with more rows, columns, and batches, the monoidal data per column and batch is still small, comprising just the count $n_x$ and the sum $s_x$.

We can express several common preprocessing operators via monoids. For instance, Standard-Scaler scales values by subtracting the mean and dividing by the standard deviation. Its monoid stores, for each column $x$ of a dataset $X$, the count $n_x$ of rows without missing value, the sum of the values in the column $s_x = \sum x$, and the sum of squares $sq_x = \sum x_i^2$. Then, transform calls from_monoid to lower these into the concrete mean and standard deviation, which it uses to scale data. The monoidal approach is not limited to preprocessing. For instance, BatchedBaggingClassifier is an ensemble where the monoid holds a list of base classifiers trained on separate batches [23]. The appendix provides the monoidal formulations of ten operators (MinMax- and StandardScaler, SimpleImputer, Project, OneHot-, Ordinal-, Target-, and HashingEncoder, SelectKBest, and Batched-BaggingClassifier) and six metrics (accuracy, $F_1$, $R^2$, DI [17], symmetric DI, and accuracy-and-DI).

**Alternatives to Monoids.** Not all machine-learning operators are monoidal, but some exhibit other properties that can also be useful towards batch-wise training. Section 3 will leverage the property of being monoidal and other alternative properties to give formal guarantees.

- An *incremental* operator has a method partial_fit : TrainableOp × BatchXy ⟶ TrainedOp. The method is designed to be invoked repeatedly on the same operator, with each call modifying the object by updating its learned coefficients. The first call changes it from trainable to trained. For example, in sklearn, MinMaxScaler and SGDClassifier have partial_fit.
- A *convergent* operator $s$ is one where, after fitting on different batches $d_1, d_2$, transforming a third batch $d_3$ yields similar results: S.fit($d_1$).transform($d_3$)≈S.fit($d_2$).transform($d_3$). Unlike being monoidal or incremental, which are algebraic properties, being convergent is an approximate property. However, deep learning universally assumes it for all layers. Likewise, stacking ensembles assume that their base predictors are convergent across folds. TargetEncoder [31] is convergent but OneHotEncoder is non-convergent. One core insight of this paper is that batch-wise training

does not require convergence; that said, for completeness, we will also show experiments that leverage convergence.

- A *pretrained* operator implements training (i.e., `fit`) as a no-op. It can be viewed as trivially incremental, monoidal, and convergent. An example is a frozen embedding such as BERT [13] if used in a pipeline without fine-tuning [9]. We also implemented HashingEncoder, a categorical encoder that lacks learned coefficients and is thus pretrained.

## 3 Task Graphs

This paper shows how to use batching to train a pipeline on a large dataset without using too much memory at any one time. The goal is to define an algorithm that takes as input a trainable pipeline and an iterator over batches of the training dataset, and outputs a trained pipeline. To train on large datasets, assuming that all operators in the pipeline implement an incremental training method `partial_fit`, we could extend the naive algorithm from Section 1 by adding an inner loop and a cache. We refer to this as the *nested-loops algorithm* and show it in Figure 3.

Unfortunately, its rigid structure of repeated full passes over all data batches causes wasteful spilling and loading. The root cause of the inflexibility is that `partial_fit` is intrinsically sequential [44]. Besides limiting scheduling flexibility and thus wastefully spilling, `partial_fit` yields a computation with a long critical path [6]. Furthermore, in cross-validation, it reduces opportunities for computation reuse [23].

```
1  for each operator S in pipeline.topological_order:
2      for each batch i ∈ 0...n − 1:
3          spill/load batches to/from cache
4          S.partial_fit(out[preds of S]ᵢ)
5      for each batch i ∈ 0...n − 1:
6          spill/load batches to/from cache
7          out[S]ᵢ ← S.transform(out[preds of S]ᵢ)
```

Figure 3: Nested-loops algorithm.

This paper introduces an alternative algorithm based on monoids and task graphs.

**Definitions**. Like a pipeline, a *task graph* is a directed acyclic graph. But unlike pipeline nodes, task graph nodes are not operators but tasks. As defined in Figure 4, each *task* (*node*) is given by an operation, a step, one or more batches, and an optional holdout. For example, the task graph in Figure 2 contains a task 'combine SIM(d0,d1,d2)', where the operation is combine, the step is SIM = SimpleImputer, and the batches are (d0,d1,d2). There are two kinds of tasks based on their *operation*: *apply* tasks (angular nodes in Figure 2) produce data and *train* tasks (rounded nodes in Figure 2) produce learned coefficients. The *step* is one of INP = input, SCR = score, or a pipeline operator. Since Figure 2 shows an example for training without cross-validation, all tasks have a *batch* belonging to the same fold (d), and none specify a *holdout*.

```
task       ::= operation step '(' batch+ ')' holdout?
operation  ::= apply | train
apply      ::= 'scan' | 'split' | 'transform' | 'predict'
train      ::= 'to_monoid' | 'combine' | 'partial_fit' | 'fit'
step       ::= 'INP' | 'SCR' | operator
batch      ::= fold idx
holdout    ::= '\' fold
```

Figure 4: Specification of a task in a task graph.

**Algorithm**. Figure 5 shows our algorithm for batch-wise training or cross-validating machine-learning pipelines. The initial phase, in L1, translates the trainable pipeline to an initial task graph, with a task scan INP($d_0$) to read the first batch ($d_0$) of the input dataset (INP) and a goal task for each operator $S$ of the pipeline. A *goal task* is a task that, when completed, contributes directly to the output of the algorithm. For example, 'combine SIM(d0,d1,d2)' in Figure 2 is

```
1  create first scan task and goal tasks
2  while any tasks ready:
3      T ← get ready task based on priority
4      spill/load batches to/from cache
5      execute T
6      if T was scan and success:
7          create next scan task
8          backward chain task creation
9  extract learned coefficients and scores
```

Figure 5: Task-graph algorithm.

a goal task that computes the learned coefficients for the SimpleImputer operator on all batches.

The main phase, in L2–L8 of Figure 5, is a worklist algorithm that executes tasks as they become ready. A task is *ready* when all its predecessors are done. L3 can pick any ready task, and can be configured with different *priorities*; for instance, to minimize spilling, our resource-aware scheduler

prioritizes tasks whose input batches are resident. L4 checks if there is enough memory for the task T, spilling batches if needed and loading any previously spilled input batches for T. L5 executes task T, performing its associated operation and storing the result with the task. L6 checks whether T was a scan task that actually returned a new batch, as opposed to having reached the end of the input iterator. In that case, L7 adds a new scan task node to the task graph for scanning the next batch of input. L8 performs backward chaining, starting from goal tasks to create their required recursive predecessors (see below for detailed rules). The final phase, in L9, reads the desired outputs from goal tasks. For example, in Figure 2, it would read the learned coefficients from 'combine SIM(d0,d1,d2)' and create a trained SimpleImputer from them in the trained pipeline.

**Backward Chaining.** The backward chaining in L8 of the algorithm from Figure 5 can be configured with different rules to obtain different regimes of training or cross-validation. We define backward chaining via rules of the form *operation* $P(d_i) \rightarrow$ *operation' $S(d_{i'})$*. The different possible matches for *operation* are defined in Figure 4, and $P, S$ match instances of *step* in the same figure. Batches $d_i, d_{i'}$ match data batches in $d_0, d_1, \ldots$. The notation $\text{preds}(S)$ refers to predecessors of step $S$ in the pipeline, plus INP for all source operators of the pipeline; furthermore, $\text{preds}(\text{SCR})$ contains the sinks of the pipeline. Below, we describe four different sets of backward chaining rules.

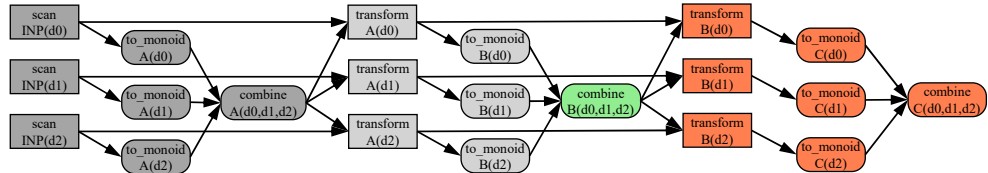

Figure 6: Task graph for training pipeline A >> B >> C on 3 batches with the full-transform regime.

*Batched training with the full-transform regime.* Our first regime works in limited memory by batching, and is faithful to the usual semantics of general (non-deep) machine learning pipelines by waiting for operators to be fully trained before applying them. Figure 6 shows an example. Let $P \in \text{preds}(S)$ and $n$ be

**R1:**        apply $P(d_i) \rightarrow$ apply' $S(d_i)$
**R2:**        apply $P(d_i) \rightarrow$ to_monoid $S(d_i)$
**R3:**     to_monoid $S(d_i) \rightarrow$ combine $S(d_0, \ldots, d_{n-1})$
**R4:** combine $S(d_0, \ldots, d_{n-1}) \rightarrow$ apply $S(d_i)$

Figure 7: Backward chaining rules for batched training with the full-transform regime.

the number of batches. Figure 7 shows the backward chaining rules (**R**). Backward chaining moves in the opposite direction of task graph edges, back from successor tasks to predecessor tasks. While there are fresh tasks, it attempts to match a fresh task against the right pattern of all rules. If the match succeeds, it finds or creates the left part indicated in the rule. For example, in Figure 6, R1 matches transform B(d0), creates transform A(d0), and creates the edge between them. The following formal guarantees (**FG**) hold for this regime. **FG1**: All schedules behave equivalently. Regardless of the schedule used by L3 of Figure 5, the final trained pipeline has the same learned coefficients, because R3 trains an operator on all batches and R4 applies such a fully-trained operator. **FG2**: This regime yields the same result as sklearn. Thanks to FG1, if suffices to show this for one schedule. Let PrioStep be a scheduling priority that favors tasks for earlier operators, or, if the operator matches, favors tasks for earlier batches. PrioStep recovers the nested-loops algorithm in Figure 3. To reduce spilling and loading, we also define PrioResourceAware that favors tasks with less non-resident input data; thanks to FG1, it is equivalent.

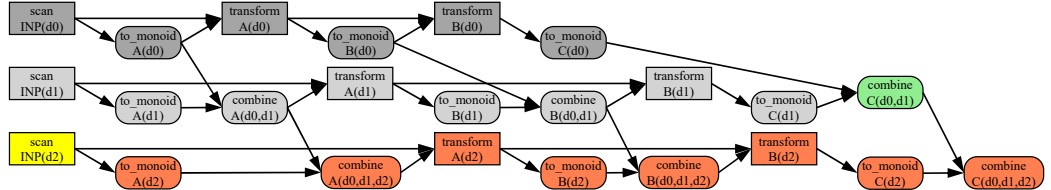

Figure 8: Task graph for training pipeline A >> B >> C on 3 batches with the partial-transform regime.

*Batched training with the partial-transform regime.* Our second regime supports incremental training by applying operators even when they are not yet fully trained. This approach is uncommon in general machine-learning but common in deep learning [27]. Figure 8 shows an example and Figure 9 shows the rules. To get the most out of the partial-transform regime,

**(R1** and **R2** from Figure 7)
**R5:**  to_monoid $S(d_0) \rightarrow$ apply $S(d_0)$
**R6:** combine $S(d_0, ..., d_i) \rightarrow$ apply $S(d_i)$
**R7:** combine $S(d_0, ..., d_i) \rightarrow$ combine $S(d_0, ..., d_{i+1})$
**R8:**  to_monoid $S(d_{i+1}) \rightarrow$ combine $S(d_0, ..., d_{i+1})$

Figure 9: Backward chaining rules for batched training with the partial-transform regime.

we define a scheduling priority PrioBatch that favors tasks for earlier batches, or if the batch matches, favors tasks for earlier steps. This regime guarantees the following: **FG3:** With PrioBatch, all operators are trained up to batch $i$ before scanning batch $i + 1$. In other words, it trains a pipeline incrementally, which can for instance be used to monitor a learning curve or for early stopping [29, 40]. Compared to the full-transform regime, the partial-transform regime is more restrictive (requiring convergent operators) and offers weaker guarantees (lacking FG1); fortunately, our approach supports both regimes.

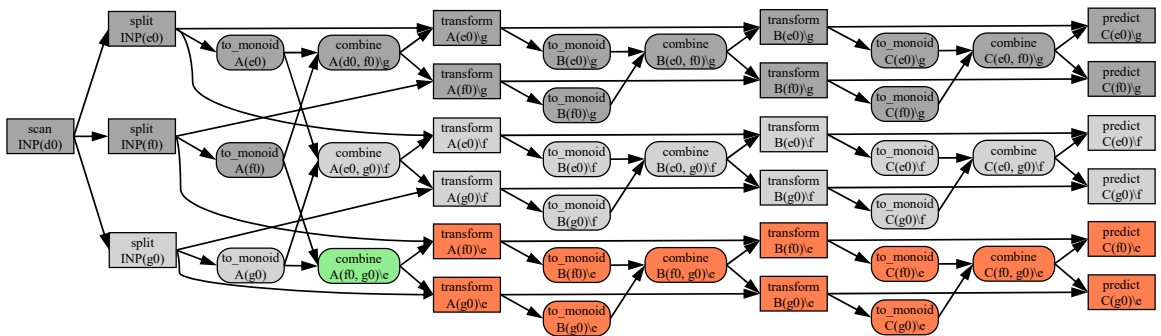

Figure 10: Cross-validation using the in-fold regime, for pipeline A >> B >> C, 3 folds, and 1 batch.

*Cross-validation with the in-fold regime.* Our third regime performs $k$-fold cross-validation with the usual semantics: for each of $k$ folds (denoted e, f, g, ...), hold out one fold $h$ and train the pipeline on the remaining $k - 1$ folds (denoted d\$h$), scoring on the held-out fold $h$. Let $f_i$ denote fold $f$ at batch index $i$, e.g., $e_0$. The resulting task graph may contain multiple different tasks 'apply $S(f_i)$' for trained copies of the same operator $S$ on the same batch $f_i$ that differ only in which fold was held out for train-

**R9:**  scan $\text{INP}(d_i) \rightarrow$ split $\text{INP}(f_i)$
**R10:**  split $\text{INP}(f_i) \rightarrow$ apply $S(f_i)\backslash h$
**R11:**  split INP $(f_i) \rightarrow$ to_monoid $S(f_i)$
**R12:** to_monoid $S(f_i) \rightarrow$ combine $S(d_i\backslash h)\backslash h$
**R13:**  apply $P(f_i)\backslash h \rightarrow$ apply $S(f_i)\backslash h$
**R14:**  apply $P(f_i)\backslash h \rightarrow$ to_monoid $S(f_i)\backslash h$
**R15:** to_monoid $S(f_i) \rightarrow$ combine $S(d_i\backslash h)\backslash h$

Figure 11: Backward chaining rules for cross-validation with the in-fold regime.

ing them. To disambiguate, let T\$h$ be a task T with a holdout fold $h$. Figure 10 shows an example and Figure 11 shows the rules. Since the total number of batches is a priori unknown and may not be a multiple of the number of folds, we must split each batch into folds when it arrives. This is handled by explicit split tasks and rule R9. Tasks for source operators ($\text{preds}(S) = \{\text{INP}\}$) can share work, since their left-hand side is still fold-agnostic, as expressed by rules R10–R12. This work sharing does not occur in non-source operators, see R13–R15. Given $k$ folds and $n$ batches, this regime makes the following formal guarantees. **FG4:** This regime yields the same result as sklearn's cross_val_score. **FG5:** A source operator has $kn$ to_monoid tasks, a non-source operator has $(k - 1)kn$ to_monoid tasks, and all operators have $(k - 1)kn$ apply tasks. It saves a factor of $k - 1$ to_monoid tasks for source operators. For space reasons, Figure 10 only illustrates the case with a single batch; see Figure 23 in the appendix for an example with multiple batches.

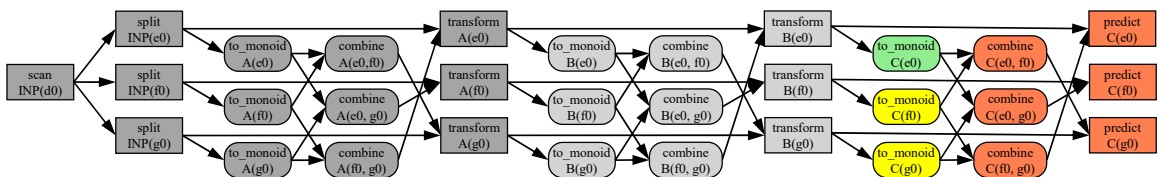

Figure 12: Cross-validation using the out-of-fold regime, for pipeline A >> B >> C, 3 folds, and 1 batch.

*Cross-validation with the out-of-fold regime.* Our fourth regime performs $k$-fold cross-validation with alternative semantics. It depends on all operators being convergent. Figure 12 shows an example and Figure 13 shows the rules. In the out-of-fold regime, training $S$ always uses only data resulting from out-of-

(**R9** from Figure 11)

**R16:** apply $P(f_i) \rightarrow$ apply $S(f_i)$

**R17:** apply $P(f_i) \rightarrow$ to_monoid $S(f_i)$

**R18:** to_monoid $S(f_i) \rightarrow$ combine $S(\mathrm{d}_i \backslash h)$ if $f \neq h$

Figure 13: Backward chaining rules for cross-validation with the out-of-fold regime.

fold applications of its predecessors $P$, and to_monoid tasks are never qualified by a holdout. This leads to more sharing. Let $k$ and $n$ be the number of folds and batches as before. **FG6**: Each operator has $kn$ to_monoid tasks and apply tasks. The out-of-fold regime unlocks the reuse enabled by monoids to save a factor of $k-1$ to_monoid and transform tasks for all operators, not just for source operators. But it lacks FG4: it may yield a different result than sklearn. For space reasons, Figure 12 only illustrates the case with 1 batch; see Figure 24 in the appendix for an example with multiple batches.

## 4 Implementation

Except for BatchedBaggingClassifier, we define all monoidal operators using a relational algebra to abstract over backends. Our algebra has three kinds of operators (c.f. Table 4 in the appendix):

- *Unary* (operators with signature Table → Table): Filter, Project, Map, OrderBy, and Alias.
- *n-ary* (operators with signature List[Table] → Table): ConcatFeatures, Join, and Scan.
- *Grouped* (operators to or from grouped data): GroupBy : Table → Grouped and Aggregate : Grouped → Table.

Our relational operators mostly follow text-book semantics, with a few deviations. As usual in dataframe systems including pandas [30] and Spark SQL [2], Grouped data is not normalized. A Table is a two-dimensional data structure with a fixed set of named columns and an ordered sequence of rows. In addition, unlike classical relational algebra, each Table has a name and an index. The *table name* is used by *n*-ary operators to identify one of their inputs, as doing so positionally would be less intuitive [3]. An *index* comprises one or more special columns of unique values that keep track of row order and identity but are separate from the payload data columns. Indexes support a common pattern in preprocessing pipelines: first preprocess different sets of columns separately (e.g., categorical encoding vs. numeric scaling), then concatenate columns belonging to the same row. To support this, each sub-pipeline must preserve row order and identity. Most of our relational operators (e.g. Map) preserve the index, and thus so do monoidal operators built with them.

Each of our relational operators has two equivalent implementations, one on pandas [30] and the other on Spark SQL [2]. Both backend libraries provide their own dataframe types, which have some differences that we needed to work around. While pandas dataframes come with an index, Spark SQL dataframes do not, so we had to implement our own index support over Spark SQL dataframes. We did that by repurposing some regular columns, and attaching metadata to each dataframe for keeping track of which columns are index vs. payload. For table names, the situation was reversed: while Spark SQL dataframes can carry a table name, pandas dataframes do not, so we had to add one as metadata. Another difference is that whereas pandas expressions execute eagerly, Spark SQL expressions lazily build up query plans for later execution. Our implementation lets Spark SQL grow these query plans across multiple tasks in a task graph and only forces execution when necessary. That gives Spark SQL's query plan optimizer the opportunity to optimize across tasks

Table 1: Memory limit at which batched succeeds but non-batched sklearn fails (RQ1).

| Dataset | Target column | Rows | Cols | Size (disk) | Memory limit |
|---------|---------------|------|------|-------------|--------------|
| KDDCup99 [15] | target | 4,898,431 | 42 | 0.8GB | 10GB |
| steam_reviews [42] | recommended | 21,747,371 | 17 | 2.2GB | 16GB |
| ecommerce_2019_oct [35] | brand | 42,448,764 | 8 | 3.2GB | 16GB |
| data_cityofchicago_taxi [11] | Trip Total | 107,780,923 | 8 | 5.7GB | 16GB |

and even across operators in a machine-learning pipeline. Our monoidal operators are oblivious of the backend, since they use our relational operators. Relational operators pick a backend by checking the type of their input data, so the same pipeline can work on both backends.

## 5 Results

This section describes our experiments to answer six research questions (RQs).

*RQ1. Can batching enable fitting pipelines on larger data without Spark SQL?* We compared our batched full-transform training regime with our pandas backend against the training of an equivalent sklearn pipeline. The pipeline comprised a SimpleImputer, OrdinalEncoder, SelectKBest, and SGDClassifier or SGDRegressor depending on the dataset. To create a limited memory setting, we controlled the memory allotted to the Python process starting from 64GB going down in steps of 2GB. We noted the value at which sklearn training fails and ran the batched training with that memory size with a batch size of 10,000. Table 1 shows the result on four large datasets. Batched training worked successfully with the memory limit, confirming that our approach enables training pipelines on datasets too large for sklearn pipelines to process. We reran the experiment on a cloud VM with only 8GB of physical memory and its default system swap space of 2GB. The result was the same: sklearn ran out of memory while our batched training succeeded.

*RQ2. When should users use the pandas backend and when the Spark SQL backend?* This experiment compared the computational performance (speed) of the pandas and Spark SQL implementations of our operators. (We also checked the predictive performance of the two backends and found them to be identical as seen in Table 9.) We record the time to train a classification pipeline on 10 OpenML [47] datasets. We upsample each dataset to observe how each of the backends performs as the data size increases. Figure 14 shows the results of this comparison. We used Spark SQL in local mode so the hardware resources are the same for the two backend runs. While neither backend is a clear winner, providing two backends enables users to choose the

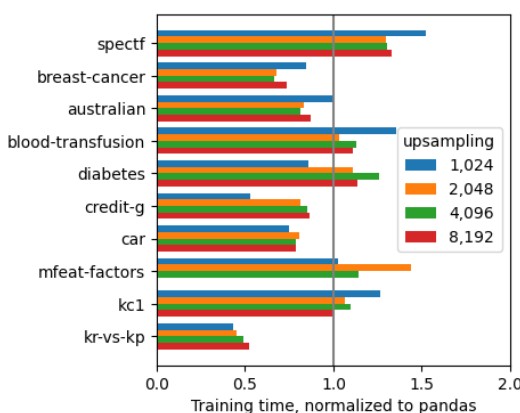

Figure 14: Ratio of training time with Spark SQL vs. pandas backend (RQ2).

faster one for their circumstances, which can obtain a substantial speedup over the other.

*RQ3. Do the pandas and Spark SQL backends yield identical results to sklearn?* We used monoids to reimplement six operators from sklearn (MinMaxScaler, StandardScaler, OneHotEncoder, OrdinalEncoder, SelectKBest, and SimpleImputer) plus two operators from scikit-learn-contrib (HashingEncoder and TargetEncoder). To check if they behave the same as the original implementation, we extracted examples from the documentation (https://scikit-learn.org/stable/modules/classes.html and http://contrib.scikit-learn.org/category_encoders/) and executed them with our implementation. We obtained the same results. In addition, we implemented tests that systematically compare the output and public state of our operators with the reference implementation. They do not always

produce equal results but very similar ones ($< 10^{-7}$ difference). Table 5 in the appendix provides a detailed summary of the experiments.

*RQ4. Does batched execution yield identical results as non-batched?* This experiment verifies that the batched task graph execution with the full-transform regime on our pandas backend generates the same outputs as a sklearn-style non-batched execution using our monoidal preprocessing operators. The pipeline for this experiment comprises a SimpleImputer, OrdinalEncoder, SelectKBest, and RandomForestClassifier. RandomForestClassifier does not support `partial_fit`, ensuring that any accuracy differences would only be due to the preprocessing prefix of the pipeline. We control the random seeds for the train-test splits. The mean accuracy and standard deviation over 5 random 80%-20% train-test splits of 10 OpenML [47] datasets was identical for the two cases (see appendix).

*RQ5. How much accuracy does partial-transform training lose?* Figure 15 shows the results from an experiment with 10 OpenML [47] datasets and 10 incremental classifiers. In each experiment, the classifier is the final operator in a pipeline with convergent preprocessing operators (SimpleImputer, MinMaxScaler, HashingEncoder). We repeated each run with 5 different random 80%-20% splits. The non-incremental regimes (non-inc-sklearn and non-inc-rasl) use a single `fit` call, and differ in whether they use sklearn preprocessing operators directly or our monoidal reimplementations. The incremental regimes

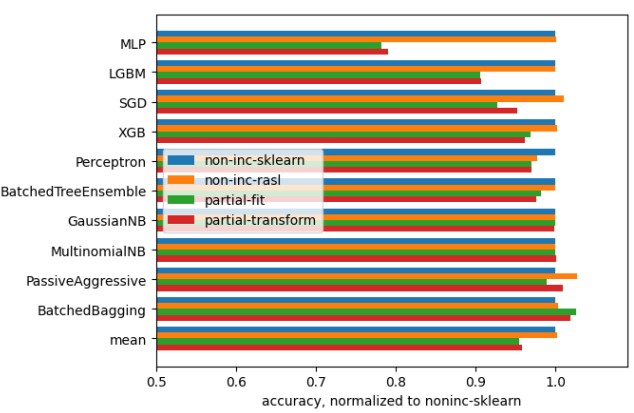

Figure 15: Accuracy normalized to non-inc-sklearn of incremental training (RQ5), averaged over 10 datasets.

(partial-fit and partial-transform) split the training set into 5 batches. Here, partial-fit refers to the case of only transforming data with preprocessing operators after they have been trained on all batches. Each bar averages the accuracy from 5 holdout splits, then normalizes to non-inc-sklearn, and finally averages over 10 datasets. While the non-incremental regimes tend to perform better (especially for MLPClassifier, where `fit` makes multiple passes over the data), the two non-incremental regimes perform similarly to each other and the two incremental regimes perform similarly to each other. For incremental training, partial-transform loses very little accuracy compared to partial-fit.

*RQ6. How effective is out-of-fold cross-validation at picking models?* To quantify how good a cross-validation regime is, we let it pick the model that has the best accuracy according to the regime, and then check where that model ranks on the holdout set. The out-of-fold regime yields a smaller task graph but also reduces isolation between training on different folds. This ranking experiment used the same 10 classifiers and 10 datasets as in RQ5. Let inf_cv$k$ and oof_cv$k$ denote the in-fold and out-of-fold cross-validation regimes, where $k$ is the number of folds, and let holdout$k$ denote batched training followed by evaluation on the holdout set, where $k$ is the number of batches. Both ranks and slow-downs are relative to holdout3 and averaged across 10 datasets, with error bars showing standard deviations; for both, lower is better. All configurations attempt 10 classifiers and the rank is computed by finding the configuration's best classifier on the rank list according to holdout3. All configurations used monoidal preprocessing operators

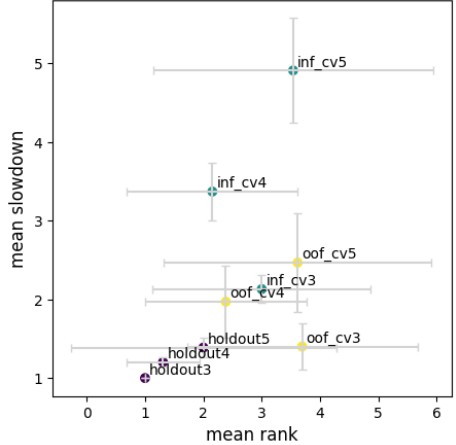

Figure 16: Correlation between mean rank and slowdown (RQ6).

but non-incremental `fit` for the final classifier. Figure 16 shows the results. Being the baseline, by definition, `holdout3` has rank 1 with zero standard deviation. Since there are 10 classifiers, random picking would yield an expected rank of 5.5. The out-of-fold regime is faster than in-fold thanks to the smaller task graph. The results for ranks are less clear due to the large error bars. Given that the out-of-fold regime may leak information between folds, it might do worse on ranking.

## 6 Related Work

We discuss prior work related to each of the three contributions of our paper listed in Section 1.

*Monoids.* Izbicki [23] defines monoidal operators including variants of Bayes classifiers, decision trees, and ensembling. He motivates these with uses for cross-validation, parallelization, and online training. Mergeable summaries [1] are monoidal approximate algorithms for frequency estimation, quantile summary, etc. In MapReduce [12], reduce operations do not have to be associative, but they often are [49], in which case they unlock more optimizations. In a streaming setting, monoids are essential to efficient incremental sliding window aggregation [20]. Steele observes that, in contrast to the linear accumulator mindset prevalent in functional programming, monoids enable binary decomposition algorithms [44]. Associative aggregation has also been at the core of data warehousing [19]. In contrast to the above-listed papers, we show how to express common sklearn operators as monoids in the context of an end-to-end machine learning pipeline.

*Task graphs.* Dask implements `partial_fit` methods to enable batching for several sklearn operators, and provides a parallel engine for "dask graphs" [39]. Like our work, Dask connects to pandas and sklearn; but unlike our work, Dask does not use monoids, nor does it tackle end-to-end pipelines. Paramo uses a task graph for cross-validation and algorithm selection of a learning pipeline on Hadoop [34]. KeystoneML provides a joint pipeline abstraction across data preparation and machine learning in Spark, with parallelism and intra-pipeline caching [43]. Helix goes one step further with inter-pipeline caching across trials [48]. Cilk defines task graphs for work stealing approach with asymptotically optimal parallel speedup [6]. Ray provides task graphs for distribution and lets users interleave training with serving [33]. In contrast to these systems, our task graphs leverage monoids for computation reuse and scheduling flexibility, contributing backward chaining rules for sklearn compatible and novel regimes.

*Machine learning and relational algebra.* We published a non-archival workshop paper on RASL [41] with an early version of most of the relational algebra operators used here, but none of the monoids built on top and no task graphs. Modin is a pandas alternative based on a dataframe algebra with a relational core [38]. We use a similar data model and similar operators, but use pandas as a backend instead of supplanting it. Several papers have melded linear algebra with relational algebra, including SparkNet [32], LaraDB [22], Weld [37], and Lara [26]. Unlike these, we use relational algebra as an intermediate layer to implement monoids usable in task graphs.

## 7 Conclusion

This paper presents techniques for training or cross-validating machine-learning pipelines in limited memory. The main idea is to process data one batch at a time, and to express operators with monoids for flexibility and reuse. Our implementation enables AutoML tools to use common sklearn APIs for end-to-end pipelines over large datasets with a single modest compute node.

## 8 Broader Impact Statement

After careful reflection, the authors have determined that this work presents no notable negative impacts to society or the environment.

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

# A Appendix

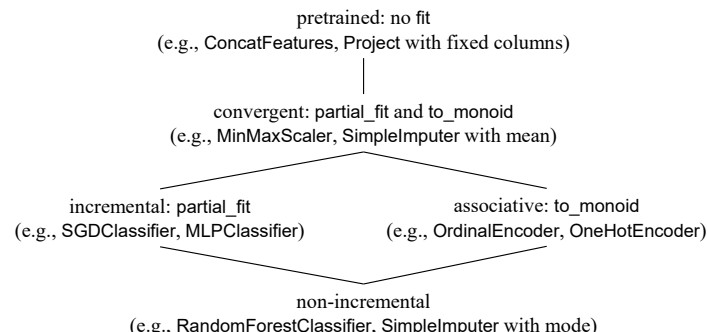

pretrained: no fit
(e.g., ConcatFeatures, Project with fixed columns)

convergent: partial_fit and to_monoid
(e.g., MinMaxScaler, SimpleImputer with mean)

incremental: partial_fit
(e.g., SGDClassifier, MLPClassifier)

associative: to_monoid
(e.g., OrdinalEncoder, OneHotEncoder)

non-incremental
(e.g., RandomForestClassifier, SimpleImputer with mode)

Figure 17: Taxonomy for operator properties from Section 2. The properites form a lattice, where each property subsumes all properties reachable by following lines "down". For instance, every convergent operator is also both incremental and associative.

Table 2: Monoids for operators. As discussed in Section 2, each associative operator provides three operations to_monoid, combine, and transform. This table specifies those operations in relational algebra; the pipe combinator **>>** denotes function composition. The from_monoid operation is inlined in transform.

| Operator | to_monoid$(X, y)$ | combine$(a, b)$ | transform$(X)$ |
|---|---|---|---|
| MinMaxScaler | $\mathrm{Agg}(lo_x : \min(x),\ hi_x : \max(x))$ | $lo_x : \min(a.lo_x, b.lo_x),$ $hi_x : \max(a.hi_x, b.hi_x)$ | $x : \dfrac{x - lo_x}{hi_x - lo_x}$ |
| StandardScaler | $\mathrm{Map}(x : x,\ sq : x^2)$ **>>** $\mathrm{Agg}(n_x : \mathrm{count}(),\ s_x : \Sigma x,\ sq_x : \Sigma sq)$ | $n_x : a.n_x + b.n_x,\ s_x : a.s_x + b.s_x,$ $sq_x : a.sq_x + b.sq_x$ | $x : \dfrac{x - s_x/n_x}{\sqrt{(sq_x - s_x^2/n_x)/n_x}}$ |
| SimpleImputer (mean) | $\mathrm{Filter}(x \neq \bot)$ **>>** $\mathrm{Agg}(n_x : \mathrm{count}(),\ s_x : \mathrm{sum}(x))$ | $n_x : a.n_x + b.n_x,\ s_x : a.s_x + b.s_x$ | $x : \mathrm{replace}(x, \{\bot : s_x/n_x\})$ |
| Project(categorical) | $\mathrm{Agg}(u_x : \mathrm{collect\_set}(x, limit))$ | $u_x : a.u_x \cup_{limit} b.u_x$ | keep $x$ for which $|u_x| \leq limit$ |
| OneHotEncoder | $\mathrm{Agg}(u_x : \mathrm{collect\_set}(x))$ | $u_x : a.u_x \cup b.u_x$ | $x_1 : \mathrm{int}(x = u_{x_1}), \ldots,$ $x_{i_{|u_x|}} : \mathrm{int}(x = u_{x_{|u_x|}})$ |
| OrdinalEncoder | $\mathrm{Agg}(u_x : \mathrm{collect\_set}(x))$ | $u_x : a.u_x \cup b.u_x$ | $x : \mathrm{replace}(x, \mathrm{encoding\_dict}(u_x))$ |
| TargetEncoder | $\mathrm{GroupBy}(x)$ **>>** $\mathrm{Agg}(n_{x_c} : \mathrm{count}(),\ s_{x_c} : \mathrm{sum}(y))$ | $n_{x_c} : a.n_{x_c} + b.n_{x_c},$ $s_{x_c} : a.s_{x_c} + b.s_{x_c},$ | $x : \mathrm{replace}(x, \{x_{x_c} : \lambda(n_{x_c}) \frac{s_{x_c}}{n_{x_c}}$ $+ (1 - \lambda(n_{x_c})) \frac{\Sigma_c\, s_{x_c}}{\Sigma_c\, n_{x_c}} \})$ |
| HashingEncoder | NoOp | NoOp | $c_i : \Sigma_x(\mathrm{hash}(x) \% N = i)$ |
| SelectKBest (f_classif) | $\mathrm{GroupBy}(y)$ **>>** $\mathrm{Agg}(n_k : \mathrm{count}(), s_{k_i} : \Sigma x_{k_i})$ $\mathrm{Map}(x_i : x_i,\ sq_i : x_i^2)$ **>>** $\mathrm{Agg}(n_i : \mathrm{count}(),\ s_i : \Sigma x_i,\ sq_i : \Sigma sq_i)$ | $n_k : a.n_k + b.n_k,\ s_{k_i} : a.s_{k_i} + b.s_{k_i},$ $n_i : a.n_i + b.n_i,\ s_i : a.s_i + b.s_i,$ $sq_i : a.sq_i + b.sq_i$ | keep $x_i$ for which $f_i$ among $k$ best where $f_i : \dfrac{\frac{\Sigma_k(s_{k_i}^2/n_k) - s_i^2/n}{|k|-1}}{\frac{sq_i - s_i^2/n - \left(\Sigma_k(s_{k_i}^2/n_k) - s_i^2/n\right)}{n-|k|}}$ and $|k|$ is the number of classes |
| BatchedBagging-Classifier(base_est) | $c : \mathrm{singleton\_list}(\mathrm{base\_est.fit}(X, y))$ | $c : \mathrm{concat\_lists}(a.c, b.c)$ | $\mathrm{mode}(c_j.\mathrm{predict}(X)$ for $c_j \in c)$ |

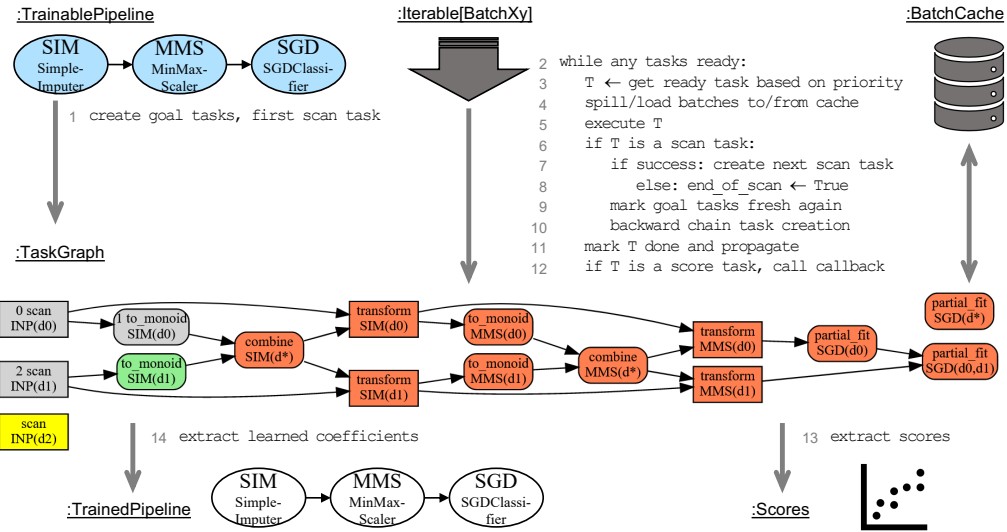

Figure 18: More detailed version of the task-graph algorithm from Figure 5, including an example input trainable pipeline, intermediate task graph, and output trained pipeline. The wildcard '*' refers to 'all batches' for situations where the total number of batches from the input 'Iterable[BatchXy]' is not yet known because the iterator has not yet been exhausted.

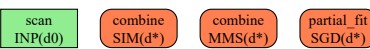

Figure 19: Initial task graph corresponding to Figure 18 L1.

Table 3: Monoids for metrics. Each monoidal metric provides three operations to_monoid, combine, and from_monoid. This table specifies those operations in relational algebra; the pipe combinator **>>** denotes function composition.

| Metric | to_monoid($y, \hat{y}, X$) | combine($a, b$) | from_monoid |
|---|---|---|---|
| accuracy | $\text{Map}(m: \text{int}(y = \hat{y}))$ **>>** $\text{Agg}(m: \Sigma m, n: \text{count}())$ | $m: a.m + b.m,\ n: a.n + b.n$ | $m/n$ |
| $F_1$ score | $\text{Map}(tp: \text{int}(\hat{y}{=}1 \wedge y{=}1),\ fp: \text{int}(\hat{y}{=}1 \wedge y{\neq}1),$ $fn: \text{int}(\hat{y}{\neq}1 \wedge y{=}1))$ **>>** $\text{Agg}(tp: \Sigma tp,\ fp: \Sigma fp,\ fn: \Sigma fn)$ | $tp: a.tp + b.tp,\ fp: a.fp + b.fp,$ $fn: a.fn + b.fn$ | $\dfrac{2tp}{2tp + fp + fn}$ |
| $R^2$ score | $\text{Map}(y: y,\ y2: y^2,\ e2: (y - \hat{y})^2)$ **>>** $\text{Agg}(n: \text{count}(),\ y: \Sigma y,\ y2: \Sigma y2,\ e2: \Sigma e2)$ | $n: a.n + b.n,\ y: a.y + b.y,$ $y2: a.y2 + b.y2,\ e2: a.e2 + b.e2$ | $1 - \dfrac{e2}{y2 - y^2/n}$ |
| disparate impact | $\text{Map}(g_{00}: \text{int}(x_p{=}0 \wedge \hat{y}{=}0),\ g_{01}: \text{int}(x_p{=}0 \wedge \hat{y}{=}1),$ $g_{10}: \text{int}(x_p{=}1 \wedge \hat{y}{=}0),\ g_{11}: \text{int}(x_p{=}1 \wedge \hat{y}{=}1))$ **>>** $\text{Agg}(g_{00}: \Sigma g_{00},\ g_{01}: \Sigma g_{01},\ g_{10}: \Sigma g_{10},\ g_{11}: \Sigma g_{11})$ | $g_{00}: a.g_{00} + b.g_{00},\ g_{01}: a.g_{01} + b.g_{01},$ $g_{10}: a.g_{10} + b.g_{10},\ g_{11}: a.g_{11} + b.g_{11}$ | $\dfrac{g_{01}/(g_{00} + g_{01})}{g_{11}/(g_{10} + g_{11})}$ |
| symmetric disparate impact | $di: \text{disp\_impact.to\_monoid}(y, \hat{y}, X)$ | $di: \text{disp\_impact.combine}(a.di, b.di)$ | $\min\left(di, \dfrac{1}{di}\right)$ |
| accuracy and disparate impact | $sdi: \text{symm\_di.to\_monoid}(y, \hat{y}, X)$ $acc: \text{accuracy.to\_monoid}(y, \hat{y}, X)$ | $sdi: \text{symm\_di.combine}(a.sdi, b.sdi)$ $acc: \text{accuracy.combine}(a.acc, b.acc)$ | $fw \cdot sdi +$ $(1 - fw) \cdot acc$ |

```
1  pipeline = (
2      (   Scan(table=it.y_true) >> Map(columns={"y": it[0]})
3        & Scan(table=it.y_pred) >> Map(columns={"p": it[0]}))
4      >> ConcatFeatures
5      >> Map(columns={"y": it.y, "p": it.p, "y2": it.y * it.y, "e2": (it.y - it.p) * (it.y - it.p)})
6      >> Aggregate(columns={"n": count(it.y), "y": sum(it.y), "y2": sum(it.y2), "e2": sum(it.e2)}))
```

Figure 20: Pipeline example for $R_2$ score. This elaborates the code of one of the metrics from Table 3. The and combinator **&** creates separate subpipelines of operators without adding edges between them.

Table 4: Relational algebra operators (c.f. Section 4). Each is implemented in an sklearn-compatible way for use in sklearn pipelines, as exemplified in Figures 20 and 21. Hyperparameters are constructor arguments.

| Operator | | Hyperparameters | | Transform | |
|----------|--|-----------------|--|-----------|--|
| Name | Description | Name | : Type | Input | → Output |
| Filter | Drop non-matched rows | pred | : List[expr] | Table | → Table |
| Project | Drop non-matched columns | columns | : Union[monoid, List[str]] | Table | → Table |
| | | drop_columns | : Union[monoid, List[str]] | | |
| Map | Assign columns, one row at a time | columns | : Dict[str, expr] | Table | → Table |
| | | remainder | : Enum["passthrough", "drop"] | | |
| ConcatFeatures | Concatenate rows | — | | List[Table] | → Table |
| Join | Combine columns of matching rows | pred | : List[expr] | List[Table] | → Table |
| | | join_type | : Enum["inner", "left", "right"] | | |
| | | name | : str | | |
| GroupBy | Create groups of rows | by | : List[expr] | Table | → Grouped |
| Aggregate | Reduce group to row | columns | : Dict[str, expr] | Grouped | → Table |
| OrderBy | Sort by columns | by | : List[expr] | Table | → Table |
| Scan | Pick out a table | table | : expr | List[Table] | → Table |
| Alias | Rename table | name | : str | Table | → Table |

```
1  prep_cats = (Project(columns=categorical(5))
2              >> SimpleImputer(strategy="constant")
3              >> HashingEncoder())
4  prep_nums = (Project(columns={"type": "number"}, dropcolumns=categorical(5))
5              >> SimpleImputer(strategy="mean")
6              >> MinMaxScaler())
7  pipeline = ((prep_cats & prep_nums) >> ConcatFeatures
8              >> SelectKBest()
9              >> MLPClassifier())
10 trained = pipeline.fit(train_X, train_y)
11 y_pred = trained.predict(test_y)
```

Figure 21: Pipeline example for multi-modal input. While this paper uses the relational algebra operators from Table 4 for implementing operators and metrics, they can also be used directly, as this example illustrates.

Table 5: Summary of tests run for RQ3. For each operator we report the number of tests and values compared to the sklearn reference implementation. The order of operations sometimes changed between the implementations, since floating-point arithmetic is not commutative, different approximations are computed. We therefore report the number of exact matches and the number of times the value are different with a difference $< 10^{-7}$. The code of the tests is available in [URL elided for double-blind review].

| Operator | Number of tests | Number of checks | Exact match | Approximate match |
|----------|-----------------|------------------|-------------|-------------------|
| MinMaxScaler | 8 | 18,052 | 8,308 | 9,744 |
| StandardScaler | 6 | 124,568 | 77,852 | 46,716 |
| OrdinalEncoder | 5 | 598,878 | 598,876 | 2 |
| SelectKBest | 4 | 144,345 | 144,267 | 78 |
| OneHotEncoder | 5 | 38,464 | 38,464 | 0 |
| SimpleImputer | 7 | 1,096,052 | 1,096,052 | 0 |
| TargetEncoder | 4 | 24,689 | 24,689 | 0 |

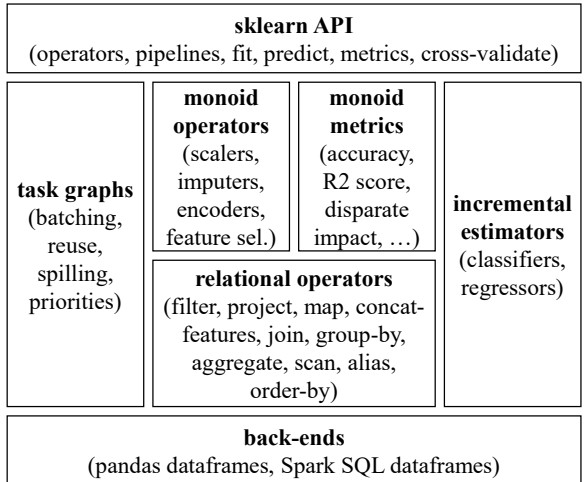

Figure 22: Architecture. Section 4 describes the implementation of our library and this figure shows it as a stack. At the base are backends for pandas and Spark SQL. Between the backends and the monoidal operators and metrics is a layer of relational algebra operators. This paper also contributes task graphs, designed to seamlessly work with existing incremental operators from sklearn and third-party sklearn-compatible libraries. At the top, our library can be consumed via an sklearn API.

Table 6: Mean accuracy (stddev) over 5 runs with vs. without batching (RQ4) on 10 OpenML datasets.

| Dataset | Rows | Cols | Full-transform | Non-batched |
|---------|------|------|----------------|-------------|
| spectf | 267 | 44 | 0.893 (0.029) | 0.893 (0.029) |
| breast-cancer | 286 | 9 | 0.724 (0.016) | 0.724 (0.016) |
| australian | 690 | 15 | 0.875 (0.018) | 0.875 (0.018) |
| blood-transfusion | 748 | 5 | 0.722 (0.018) | 0.722 (0.018) |
| diabetes | 768 | 9 | 0.759 (0.017) | 0.759 (0.017) |
| credit-g | 1,000 | 20 | 0.760 (0.011) | 0.760 (0.011) |
| car | 1,728 | 7 | 0.960 (0.007) | 0.960 (0.007) |
| mfeat-factors | 2,000 | 217 | 0.967 (0.005) | 0.967 (0.005) |
| kc1 | 2,109 | 22 | 0.854 (0.014) | 0.854 (0.014) |
| kr-vs-kp | 3,196 | 37 | 0.990 (0.002) | 0.990 (0.002) |

Table 7: Incremental classifiers used in RQ5 and RQ6. All of these either already came with a `partial_fit` method or we added it where missing and distribute it with our library.

| Library | Operator |
|---------|----------|
| sklearn [8] | GaussianNB |
| sklearn [8] | MultinomialNB |
| sklearn [8] | Perceptron |
| sklearn [8] | SGDClassifier |
| sklearn [8] | PassiveAggressiveClassifier |
| sklearn [8] | MLPClassifier |
| Lale [4] | BatchedBaggingClassifier |
| Snap ML [14] | BatchedTreeEnsembleClassifier |
| LightGBM [24] | LGBMClassifier |
| XGBoost [10] | XGBClassifier |

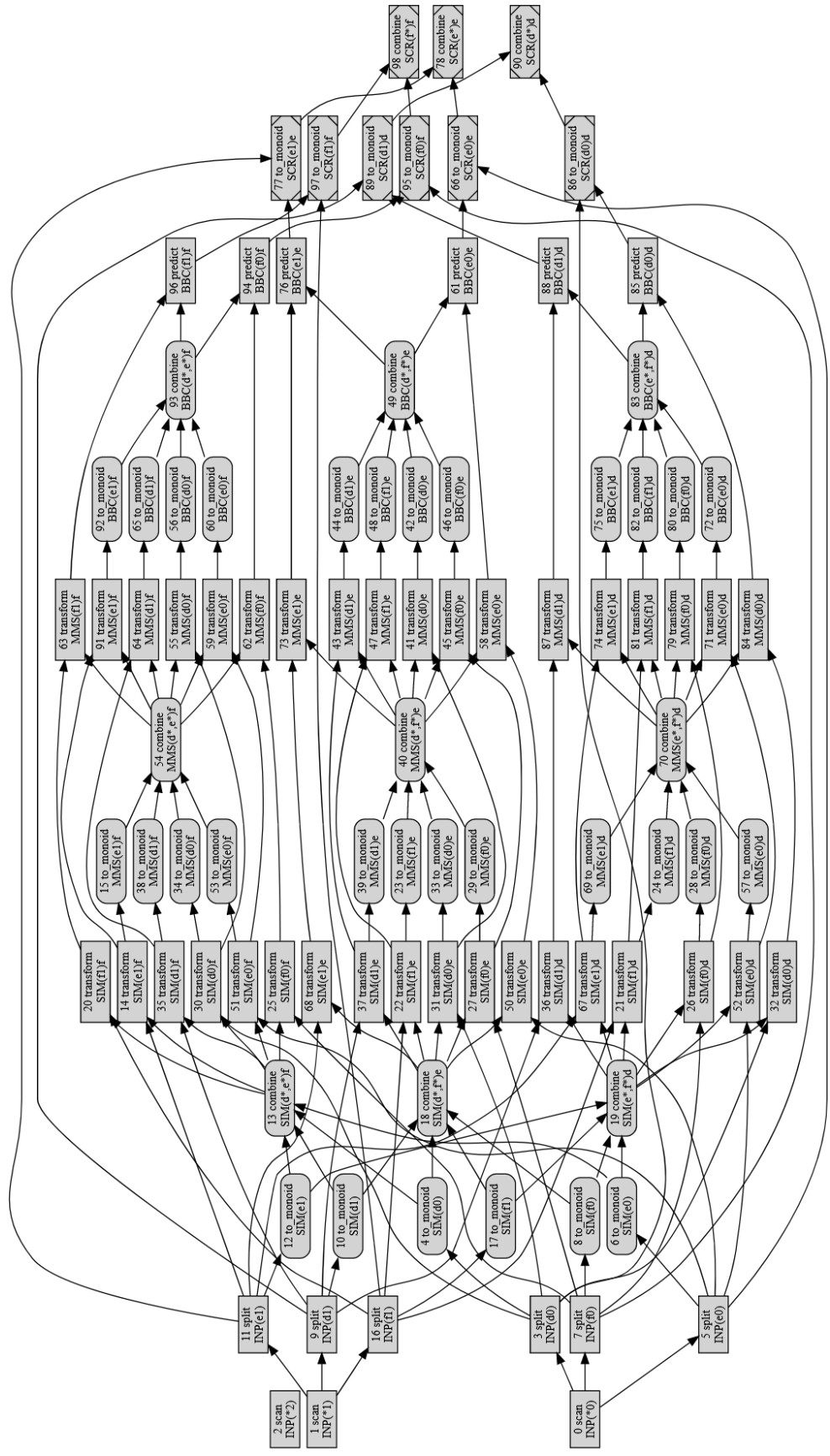

Figure 23: Task graph for cross-validation using the in-fold regime with 3 folds (d,e,f) and 2 batches (0,1). The pipeline is SIM >> MMS >> BBC, where SIM is SimpleImputer, MMS is MinMaxScaler, and BBC is BatchedBaggingClassifier. All three of these operators are monoidal. This figure also shows score tasks denoted SCR using our monoidal implementation of the accuracy metric.

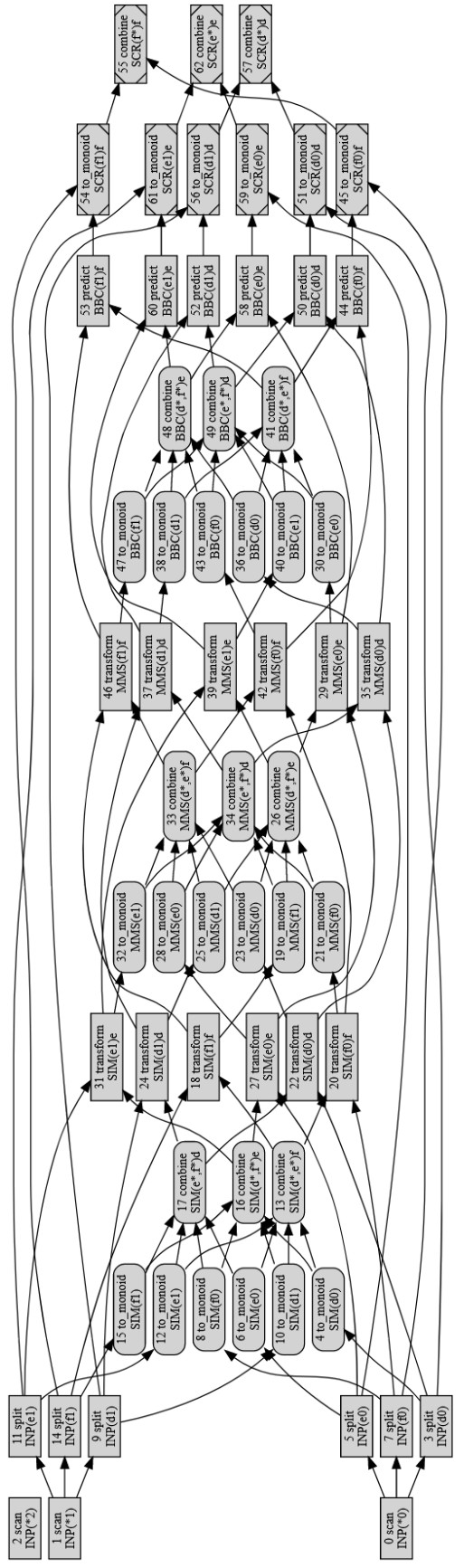

Figure 24: Task graph for cross-validation using the out-of-fold regime with 3 folds (d,e,f) and 2 batches (0,1). The pipeline is `SIM >> MMS >> BBC`, where `SIM` is `SimpleImputer`, `MMS` is `MinMaxScaler`, and `BBC` is `BatchedBaggingClassifier`. All three of these operators are monoidal. This figure also shows score tasks denoted `SCR` using our monoidal implementation of the accuracy metric.

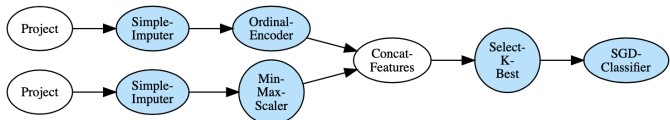

Figure 25: Pipeline used for comparison of batched training to sklearn training (RQ1). SGDClassifier is replaced with SGDRegressor for regression tasks. Operators shown in white are pretrained, whereas blue indicates associative or incremental.

Table 8: OpenML [47] datasets used in RQ5 and RQ6.

| Dataset | Rows | Cols |
| --- | ---: | ---: |
| blood-transfusion | 748 | 5 |
| diabetes | 768 | 9 |
| credit-g | 1,000 | 20 |
| car | 1,728 | 7 |
| mfeat-factors | 2,000 | 217 |
| kc1 | 2,109 | 22 |
| kr-vs-kp | 3,196 | 37 |
| sylvine | 5,124 | 21 |
| phoneme | 5,404 | 6 |
| jungle-chess | 44,819 | 7 |

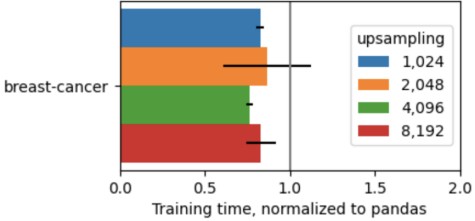

Figure 26: Ratio of training time with Spark SQL vs. pandas backend (RQ2) with error bars.

Table 9: Raw results for RQ2.

| dataset | setting | fraction | holdout_accuracy_mean | holdout_accuracy_stddev | time_mean | time_stddev | ratio (spark/pandas) |
|---|---|---|---|---|---|---|---|
| spectf | spark | 1024 | 0.698275862 | 0 | 62.33600473 | 0 | 1.51992169 |
| spectf | pandas | 1024 | 0.698275862 | 0 | 41.01264238 | 0 | 1 |
| spectf | spark | 2048 | 0.698275862 | 0 | 131.1390002 | 0 | 1.297772454 |
| spectf | pandas | 2048 | 0.698275862 | 0 | 101.0493016 | 0 | 1 |
| spectf | spark | 4096 | 0.698275862 | 0 | 207.6392157 | 0 | 1.306535496 |
| spectf | pandas | 4096 | 0.698275862 | 0 | 158.9235168 | 0 | 1 |
| spectf | spark | 8192 | 0.698275862 | 0 | 452.0865004 | 0 | 1.328498656 |
| spectf | pandas | 8192 | 0.698275862 | 0 | 340.2988014 | 0 | 1 |
| breast-cancer | spark | 1024 | 0.715789474 | 0 | 11.77873111 | 0 | 0.846154598 |
| breast-cancer | pandas | 1024 | 0.715789474 | 0 | 13.92030621 | 0 | 1 |
| breast-cancer | spark | 2048 | 0.715789474 | 0 | 17.53592038 | 0 | 0.677604551 |
| breast-cancer | pandas | 2048 | 0.715789474 | 0 | 25.87928367 | 0 | 1 |
| breast-cancer | spark | 4096 | 0.715789474 | 0 | 39.06328797 | 0 | 0.665626869 |
| breast-cancer | pandas | 4096 | 0.715789474 | 0 | 58.68646502 | 0 | 1 |
| breast-cancer | spark | 8192 | 0.715789474 | 0 | 96.74211335 | 0 | 0.735493191 |
| breast-cancer | pandas | 8192 | 0.715789474 | 0 | 131.5336628 | 0 | 1 |
| australian | spark | 1024 | 0.600877193 | 0 | 39.93376708 | 0 | 0.998017721 |
| australian | pandas | 1024 | 0.600877193 | 0 | 40.01308417 | 0 | 1 |
| australian | spark | 2048 | 0.600877193 | 0 | 89.66839337 | 0 | 0.832317072 |
| australian | pandas | 2048 | 0.600877193 | 0 | 107.7334545 | 0 | 1 |
| australian | spark | 4096 | 0.600877193 | 0 | 196.3755603 | 0 | 0.812198922 |
| australian | pandas | 4096 | 0.600877193 | 0 | 241.7825916 | 0 | 1 |
| australian | spark | 8192 | 0.600877193 | 0 | 451.3203506 | 0 | 0.872523768 |
| australian | pandas | 8192 | 0.600877193 | 0 | 517.2585175 | 0 | 1 |
| blood-transfusion | spark | 1024 | 0.732793522 | 0 | 45.32721543 | 0 | 1.354819502 |
| blood-transfusion | pandas | 1024 | 0.732793522 | 0 | 33.45627618 | 0 | 1 |
| blood-transfusion | spark | 2048 | 0.732793522 | 0 | 56.06554794 | 0 | 1.030880564 |
| blood-transfusion | pandas | 2048 | 0.732793522 | 0 | 54.38607526 | 0 | 1 |
| blood-transfusion | spark | 4096 | 0.732793522 | 0 | 152.1027329 | 0 | 1.131169147 |
| blood-transfusion | pandas | 4096 | 0.732793522 | 0 | 134.465065 | 0 | 1 |
| blood-transfusion | spark | 8192 | 0.732793522 | 0 | 349.2777286 | 0 | 1.112427673 |
| blood-transfusion | pandas | 8192 | 0.732793522 | 0 | 313.9779215 | 0 | 1 |
| diabetes | spark | 1024 | 0.661417323 | 0 | 44.60869765 | 0 | 0.856111892 |
| diabetes | pandas | 1024 | 0.661417323 | 0 | 52.10615349 | 0 | 1 |
| diabetes | spark | 2048 | 0.661417323 | 0 | 87.30375195 | 0 | 1.113078156 |
| diabetes | pandas | 2048 | 0.661417323 | 0 | 78.43452096 | 0 | 1 |
| diabetes | spark | 4096 | 0.661417323 | 0 | 245.0170028 | 0 | 1.260857635 |
| diabetes | pandas | 4096 | 0.661417323 | 0 | 194.3256686 | 0 | 1 |
| diabetes | spark | 8192 | 0.661417323 | 0 | 482.963865 | 0 | 1.136070077 |
| diabetes | pandas | 8192 | 0.661417323 | 0 | 425.1180229 | 0 | 1 |
| credit-g | spark | 1024 | 0.693939394 | 0 | 63.11257815 | 0 | 0.527458551 |
| credit-g | pandas | 1024 | 0.693939394 | 0 | 119.6540999 | 0 | 1 |
| credit-g | spark | 2048 | 0.693939394 | 0 | 159.8480685 | 0 | 0.81083415 |
| credit-g | pandas | 2048 | 0.693939394 | 0 | 197.1402764 | 0 | 1 |
| credit-g | spark | 4096 | 0.693939394 | 0 | 364.5939066 | 0 | 0.853200242 |
| credit-g | pandas | 4096 | 0.693939394 | 0 | 427.3251326 | 0 | 1 |
| credit-g | spark | 8192 | 0.693939394 | 0 | 992.3586102 | 0 | 0.86239563 |
| credit-g | pandas | 8192 | 0.693939394 | 0 | 1150.699952 | 0 | 1 |
| car | spark | 1024 | 0.695271454 | 0 | 46.3204205 | 0 | 0.7487929 |
| car | pandas | 1024 | 0.695271454 | 0 | 61.86012244 | 0 | 1 |
| car | spark | 2048 | 0.695271454 | 0 | 111.5793927 | 0 | 0.807539845 |
| car | pandas | 2048 | 0.695271454 | 0 | 138.1719966 | 0 | 1 |
| car | spark | 4096 | 0.695271454 | 0 | 223.1127155 | 0 | 0.789894795 |
| car | pandas | 4096 | 0.695271454 | 0 | 282.4587741 | 0 | 1 |
| car | spark | 8192 | 0.695271454 | 0 | 480.9890895 | 0 | 0.787069439 |
| car | pandas | 8192 | 0.695271454 | 0 | 611.1139195 | 0 | 1 |
| mfeat-factors | spark | 1024 | 0.090909091 | 0 | 1592.317333 | 0 | 1.026821316 |
| mfeat-factors | pandas | 1024 | 0.090909091 | 0 | 1550.724851 | 0 | 1 |
| mfeat-factors | spark | 2048 | 0.096969697 | 0 | 2660.711861 | 0 | 1.438332237 |
| mfeat-factors | pandas | 2048 | 0.096969697 | 0 | 1849.85902 | 0 | 1 |
| mfeat-factors | spark | 4096 | 0.081818182 | 0 | 9919.777119 | 0 | 1.144714836 |
| mfeat-factors | pandas | 4096 | 0.081818182 | 0 | 8665.719014 | 0 | 1 |
| mfeat-factors | spark | 8192 | -1 | 0 | -1 | 0 | -1 |
| mfeat-factors | pandas | 8192 | -1 | 0 | -1 | 0 | -1 |
| kc1 | spark | 1024 | 0.844827586 | 0 | 224.4481311 | 0 | 1.266899141 |
| kc1 | pandas | 1024 | 0.844827586 | 0 | 177.1633778 | 0 | 1 |
| kc1 | spark | 2048 | 0.844827586 | 0 | 484.9539335 | 0 | 1.063267111 |
| kc1 | pandas | 2048 | 0.844827586 | 0 | 456.097935 | 0 | 1 |
| kc1 | spark | 4096 | 0.844827586 | 0 | 1478.12337 | 0 | 1.093927815 |
| kc1 | pandas | 4096 | 0.844827586 | 0 | 1351.20741 | 0 | 1 |
| kc1 | spark | 8192 | 0.844827586 | 0 | 5471.855341 | 0 | 0.993253763 |
| kc1 | pandas | 8192 | 0.844827586 | 0 | 5509.020497 | 0 | 1 |
| kr-vs-kp | spark | 1024 | 0.527014218 | 0 | 200.2625384 | 0 | 0.435140871 |
| kr-vs-kp | pandas | 1024 | 0.527014218 | 0 | 460.2246115 | 0 | 1 |
| kr-vs-kp | spark | 2048 | 0.527962085 | 0 | 430.8784208 | 0 | 0.452880949 |
| kr-vs-kp | pandas | 2048 | 0.527962085 | 0 | 951.4165287 | 0 | 1 |
| kr-vs-kp | spark | 4096 | 0.527962085 | 0 | 1025.327973 | 0 | 0.492221551 |
| kr-vs-kp | pandas | 4096 | 0.527962085 | 0 | 2083.061925 | 0 | 1 |
| kr-vs-kp | spark | 8192 | 0.527962085 | 0 | 2324.238755 | 0 | 0.525030237 |
| kr-vs-kp | pandas | 8192 | 0.527962085 | 0 | 4426.866473 | 0 | 1 |

Table 10: RQ2 updated results for breast cancer averaged over 5 runs.

| dataset | setting | fraction | holdout_accuracy_mean | holdout_accuracy_stddev | time_mean | time_stddev | ratio_mean (spark/pandas) | ratio_stddev (spark/pandas) |
|---|---|---|---|---|---|---|---|---|
| breast-cancer | spark | 1024 | 0.72 | 0.064288133 | 13.27299833 | 0.793947706 | 0.825600297 | 0.023497824 |
| breast-cancer | pandas | 1024 | 0.72 | 0.064288133 | 16.06694379 | 0.562293044 | 0.825600297 | 0.023497824 |
| breast-cancer | spark | 2048 | 0.72 | 0.064288133 | 29.41563063 | 8.315909176 | 0.86576034 | 0.256408931 |
| breast-cancer | pandas | 2048 | 0.72 | 0.064288133 | 34.05961185 | 0.801125363 | 0.86576034 | 0.256408931 |
| breast-cancer | spark | 4096 | 0.72 | 0.064288133 | 58.96682754 | 2.027603695 | 0.763718999 | 0.018693966 |
| breast-cancer | pandas | 4096 | 0.72 | 0.064288133 | 77.2890501 | 4.404486342 | 0.763718999 | 0.018693966 |
| breast-cancer | spark | 8192 | 0.72 | 0.064288133 | 146.1375819 | 16.7489225 | 0.831358467 | 0.085012886 |
| breast-cancer | pandas | 8192 | 0.72 | 0.064288133 | 175.6816947 | 5.799947915 | 0.831358467 | 0.085012886 |

