# OpenReview forum: "Training and Cross-Validating Machine Learning Pipelines with Limited Memory"
_automl.cc/AutoML/2024/Conference — AutoML 2024_

### Official Review · Reviewer_wMqp · 2024-03-17

**Potential Impact On The Field Of Automl Rating:** 3
**Technical Quality And Correctness Rating:** 2
**Clarity Rating:** 3

**Summary Of Contributions:**

The main contributions of the paper are (1) a theoretical framework based on monoids to represent the (training) operations for a collection of preprocessing methods, (2) one batched algorithm with four formal task graphs based on the framework mentioned above to optimize scheduling w.r.t. memory, (3) an open-source implementation of the algorithm, and (4) results showcasing their algorithm and the different task graphs.

**Actions Required To Increase Overall Recommendation:**

Overhauling parts of the presentation of the experiments, as outlined above, would be beneficial.

Moreover, if time permits, replacing/improving RQ6 would convince me to increase my recommendation score.

**Clarity:**

* Clarity could be improved in line 37 by mentioning that out is a collection/map/list of elements.
* Line 88 introduces Figure 2. It might be helpful to mention that this figure is also about imputing missing values with the mean at that point (if space constraints allow for it).
* Line 120 "stacking ensembles assume it for their base predictors.": This needs to be clarified. Do the authors understand it as batches equal to folds?  * It might be worthwhile to replace algorithms/procedures with latex algorithms instead of using figures (e.g., Figure 5) to improve clarity and readability.
* Table 1 is very difficult to read; if more space is available, a different table structure might be advised.
* Line 310, the referenced sklearn examples are not referencing any specific examples? So which were used in the end?
* The appendix provides a lot of useful information, but the main paper does not. I recommend adding such pointers/references.
* In RQ5, the difference between partial fit and transform is unclear to me. Is partial transform using a full fit, but does preprocessing only in batches?
* It would be clearer to mention immediately instead of after the results that cross-validation with the out-of-fold regime yields information leakage during cross-validation.
* To me, it seems that the focus on monoids for preprocessing and batching, which is present in the paper, was lost in the title and abstract. It might be worth re-evaluating if the title and abstract do not overstate the contribution to clarify the scope of the paper (as the model part has not been touched).

**Overall Review:**

The positive aspects of the paper are: (1) a sound and strong theoretical framework that is worth sharing with the community, (2) an open-source and readily usable implementation of the framework that is reliable, as shown in RQ1-RQ4, (3) contributions to an important research direction for making (Auto)ML more accessible.

The negative aspects of the paper are: (1) some clarity issues as outlined above and (2) weak and imprecise experiments compared to what could have been possible to verify the worth of the contributions.

**Potential Impact On The Field Of Automl:**

The paper is impactful. Several AutoML systems (and ML systems in general) might switch to the proposed algorithms or implementations to optimize their systems for low-memory or large-data scenarios. Moreover, the proposed framework offers a solid foundation for future work and adaptation to other use cases.

**Reproducibility:**

From going through the code, I believe that the code is well structured, has a good readme and seems to be reproducible.

**Review Confidence:**

4

**Review Rating:**

8

**Review Summary:**

In summary, the paper has a solid theoretical foundation but weak experiments.

Yet, I believe that the experiments (if given more space for their presentation and some additional thought for the setup) would reach the same conclusion while being much more convincing.

Hence, I recommend accepting the paper, as most of my concerns can be resolved and are not about the proposed contribution itself.

**Technical Quality And Correctness:**

I believe the proposed framework, algorithms, task graphs, and implementation to be correct and of high quality (Sections 2, 3, 4). However, I have some specific concerns regarding the cross-validation usage and the experiments.


# Cross-Validation Usage

Figure 8 and the text in line 215 ff. imply that the proposed cross-validation algorithm performs cross-validation per batch d0 and not across all data.   In other words, this does not do batched cross-validation but rather batches of cross-validation. What is the intended usage here? Depending on the usage, the implications for memory consumption are very different.   Furthermore, it is unclear in the experiments if this plays a role or if the entire training data was always one batch when using cross-validation (RQ6).

# Experiments

## RQ1
For RQ1, it is unclear which component of the pipeline ran out of memory here. Did the preprocessing not fit into memory, or did SGD require too much memory during training? Also, how many batches were used?

## RQ2

* Why is the 8192 upsampling bar missing for mfeat-factors?
* Line 295: "The predictive performance [...] to be identical." - please provide these numbers in the appendix so the reader can also judge this.

### Time Measurements

As far as I can tell, this experiment did not repeat the training time measurements. It is unclear how much the noise from the OS and system load would influence these measurements. Generally, repeating time measurements due to such noise is critical.    What are the absolute time differences?

If the absolute time of pandas is just seconds, the time differences displayed here are insignificant and potentially just noise.  It would be worthwhile to consider larger datasets to answer RQ2. It seems to me right now that the answer to RQ2 might be entirely an artifact of noise and/or insignificant differences.

## RQ3
Please provide the numbers of the results in the appendix.

## RQ6

* Which model was trained in the baseline/holdout? As far as I can tell, only one model was fit for holdout. If so,  the plot compares model selection with the potential to overfit (the CV approaches) to one random sample (the holdout baseline).
* Are the models doing partial fits for the holdout baseline? If so, are we comparing with CV without partial fits?
* Does this experiment use any repeats per dataset?
* It is concerning that holdout performs best; could this be an artifact of overfitting for the CV-based approaches or not comparing apples to apples?

 ### Goal of this RQ

I am unsure about the merit of this research question. In line with the previous research questions, it would have been more insightful to analyze whether the inf_cv recovers sklearn's cross-validation score and (as expected due to leakage) oof_cv does not while being faster.

This RQ and its results confuse me more than they provide answers to a problem within the paper/contribution.

## General
I found some of the choices that differ between research questions very confusing.

* The authors used three repeats for RQ5, but five for RQ4. On this note, why was holdout instead of cross-validation used at all?
* In RQ5, stratified 90-10 splits are used, but in RQ4, 80-20 splits are used without stating whether they were stratified or not.
* In RQ5, balanced accuracy was used, but in RQ4, accuracy.

 ### Other

* The paper provides the monoid framework for an almost arbitrary set of preprocessing methods and metrics. Yet, this set is likely not arbitrary because other preprocessing methods and metrics might not be expressable with monoids. I suggest including a discussion about potential methods and metrics that can be expressed within the framework and explicitly mentioning commonly used methods that are not expressable in this way. This limitation of the contribution needs to be discussed in more detail.
* Regarding Line 203 ff., "This approach is uncommon in general machine-learning but common in deep learning": The given relation to deep learning is overstated and needs some clarification. While it is true that, in essence, we have incremental operations during deep learning, the type of approach of this paper (preprocessing) is also often done very similarly for deep learning approaches. E.g., for deep learning, when using tabular data, we still commonly apply to preprocess non-incrementally. Likewise, if we have some channel or color preprocessing for images, this is also employed non-incrementally. Right now, the statements imply that what the authors are doing here for non-deep learning (non-incremental operations during preprocessing) is common in deep learning. Incremental operations within the model are common in deep learning but not during preprocessing, model validation, and scoring.

---

### Official Review · Reviewer_LxCE · 2024-03-25

**Potential Impact On The Field Of Automl Rating:** 3
**Technical Quality And Correctness Rating:** 3
**Clarity Rating:** 4

**Summary Of Contributions:**

The authors address the challenge of training and cross-validating machine learning pipelines on large datasets while limited by memory resources. By leveraging monoids and task graphs, the authors propose a method allowing efficient batch-wise data processing, thereby reducing memory overhead without significantly sacrificing performance. The approach is compatible with the scikit-learn library, making it broadly usable for the AutoML community. Experiments aim to demonstrate the method's effectiveness from various angles, showing its ability to handle large datasets on low-resource hardware setups.

**Actions Required To Increase Overall Recommendation:**

- Include a more comprehensive comparison with state-of-the-art methods focusing on efficiency and effectiveness.
- Explain further why batching can be associated with monoids, emphasizing the associative properties and identity elements.
- How does the method scale to even larger datasets, where spilling on the disk might become frequent?

**Clarity:**

The paper is well-organized, with clear sections, from the introduction and the problem setting to the proposed approach, implementation details, and experimental evaluation, including structured answers to the proposed research questions. The writing style is suitably easy to follow.

**Overall Review:**

Pros:
- The concept of monoids, originating from abstract algebra, is applied innovatively in the context of machine learning pipelines to enable batch-wise data processing. The associative property of monoids ensures that the final aggregation of batch results is consistent and that distributed computational resources can be leveraged efficiently due to parallelization.
- The paper is well-organized and has a clear and concise writing style.
- Ten OpenML datasets were used for evaluation. The implications of the Pandas and Spark SQL backends and partial-transform training regarding performance were sufficiently evaluated.

Cons:
- A more direct comparison with state-of-the-art methods in terms of efficiency and accuracy could further highlight the contributions of this work.
- The paper asserts compatibility with the scikit-learn API. However, detailed insights into the integration process with existing AutoML frameworks are limited. The same holds for other types of data-preprocessing operators.

**Potential Impact On The Field Of Automl:**

Combining a monoidal framework with task graphs for batch processing and cross-validation could broaden the accessibility of AutoML to a broader range of users, including those with constrained computational budgets. The approach could also enhance the scalability and efficiency of AutoML systems.

**Review Confidence:**

4

**Review Rating:**

8

**Review Summary:**

The paper presents a novel and technically sound approach to address the challenge of training and cross-validating machine learning pipelines on large datasets within the constraints of limited memory resources. By applying monoidal frameworks alongside task graphs for efficient batch-wise data processing, the authors offer a solution that mitigates memory overhead and retains compatibility with widely used ML libraries, thus ensuring applicability within the AutoML community. The experimental evaluation is sufficient to support the author's claims regarding performance. All in all, regarding the effectiveness and relevance of the approach, I recommend accepting the paper.

**Technical Quality And Correctness:**

The technical quality of this paper is sound, supported by algebraic theory and experiments. The authors present their innovative approach effectively. However, the paper might benefit from proof of why batching can be associated with a monoid, including associative operators and an identity element.

---

### Official Review · Reviewer_ZRYr · 2024-04-02

**Potential Impact On The Field Of Automl Rating:** 2
**Technical Quality And Correctness Rating:** 3
**Clarity Rating:** 2

**Summary Of Contributions:**

The work discusses how to use monoids to allow batch training of classical machine learning pipelines. Different modes are discussed of how the proposed framework can be implemented and how those relate to the classical sklearn pipelines. The performance, in terms of reduced memory requirements, is validated on a few tasks, showing that the proposed framework is capable of drastically reducing memory requirements while largely preserving model accuracy.

**Actions Required To Increase Overall Recommendation:**

As suggested in the clarity section, the work should provide more clarification and explanation for the parts that are further empirically analyzed while moving other parts to the appendix. Otherwise the paper is very difficult to follow.

**Clarity:**

The paper is very dense and often very difficult to follow as it is a dense text interspersed with figures aimed at supporting the text. The work could benefit from moving the backwards chaining rules that are not empirically validated to the appendix to leave more room for discussion and explanation for those methods that are further empirically analyzed.

**Overall Review:**

The paper proposes an interesting change to classical ML pipelines that could be beneficial to the larger AutoML community, besides allowing for memory-efficient cross-validation. For example, the work might also have stronger implications for multi-fidelity AutoML, though the work does not discuss any AutoML related work.

**Potential Impact On The Field Of Automl:**

I do believe the work could have a medium impact as it shows how AutoML can benefit from Monoids. However, the impact is likely more limited as the work only shows it for cross-validation as a stand-alone without using it in an AutoML pipeline or framework.

**Review Confidence:**

3

**Review Rating:**

7

**Review Summary:**

A very dense paper, that is difficult to follow but could be of interest to the broader AutoML community.

**Technical Quality And Correctness:**

The work was at times difficult to follow but I do believe that it is of high technical quality, though I might have overlooked something.

---

### Meta-Review · Area_Chair_ou2H · 2024-04-20

**Paper Recommendation:** Accept
**Confidence:** 4

**Metareview:**

When it comes to the quality of the work, the reviewers seem to be in agreement.

However, I would like to raise the following important point. AutoML is one of the conferences that sets a clear standard when it comes to reproducibility and code availability (see the call for papers). Open science is not optional. The reproducibility reviewer has flagged that this paper is not accompanied by the proper code. In their reply, the authors stated that they would release their code upon the acceptance of the paper. By adhering to such (closed science) practice, they have bypassed the reproducibility review process, and not offered the reproducibility reviewer the chance to review this paper. While clearly stated by the reproducibility reviewer, the authors did not correct this during the discussion phase. The call for papers also clearly points towards the anonymized Git platform anon-github, further facilitating and explaining the emphasis on open science.

As such, based on the call for papers, this paper can not be accepted to the main track of the AutoML conference, but should rather be submitted to the workshop track. However, I would also suggest take into consideration the following matters:
*) The current scientific system is unfortunately not yet fully geared towards open science, and at many other conferences authors get away by keeping their code and datasets closed
*) The authors do state that they intend to release the code as part of the camera-ready copy
*) This situation could be a genuine misunderstanding because of the ambitious (and correct) goals that the AutoML conference sets

I am willing to give the authors the benefit of the doubt, and will advice to accept this paper for the main track of the conference under the absolute condition that the code will be made available as part of the library for the CRC copy, as they promise in the paper. This does have the unfortunate consequence that the reproducibility review has been bypassed, and as such I would impress upon the authors to be more responsible on future occasions and follow the call for papers.

===== Reference to the call for papers of the main track: =====
> We strongly value reproducibility as an integral part of scientific quality assurance. Therefore, we require that all submissions are accompanied by a link to an open-source repository providing an implementation (if empirical results are part of the paper).

===== Reference to the call for papers for the workshop track: =====
> We strongly encourage you also to submit the reproducibility checklist and open up your source code and data. However, this is not required for this track. Nevertheless, we will give reproducibility badges to all accepted submissions with a reproducibility checklist, open-source code and open-access data.

---

### Decision · Program_Chairs · 2024-04-29

**Decision:**

Accept

**Comment:**

After careful consideration and in-depth discussions, the PCs and GC follow the recommendation of the AC and accept the paper to the method track. We nevertheless strongly emphasize that this is an exceptional decision because of the partly missing source code for reproducing the results. The acceptance to the main track is, therefore, conditional on the fact that the authors have to provide the full source code for the CRC deadline on August 8th. If the authors will not provide it, the paper will be removed from the program and will not be part of the conference proceedings. We further highlight that this meant as a precedent, and in future iterations of the conference, the PCs and GC might decide otherwise under such circumstances.